# Translation and Cross-Cultural Adaptation of the International Questionnaire to Measure the Use of Complementary and Alternative Medicine (I-CAM-Q) for the Polish and Cross-Sectional Study

**DOI:** 10.3390/ijerph20010124

**Published:** 2022-12-22

**Authors:** Aneta Brygida Jędrzejewska, Barbara Janina Ślusarska, Krzysztof Jurek, Grzegorz Józef Nowicki

**Affiliations:** 1Department of Family and Geriatric Nursing, Medical University of Lublin, Staszica 6 Str., PL-20-081 Lublin, Poland; 2Institute of Sociological Sciences, The John Paul II Catholic University of Lublin, Al. Racławickie 14 Str., PL-20-950 Lublin, Poland

**Keywords:** complementary and alternative medicine, questionnaire I-CAM-Q, linguistic and intercultural adaptation, cross-sectional study

## Abstract

According to the World Health Organization (WHO), complementary and alternative medicine (CAM) encompasses a broad set of health care practices that are not part of a country’s traditional or conventional medicine and are not fully integrated into the prevailing health care system. The aim of this study is the linguistic and cross-cultural adaptation of the Polish version of the International Questionnaire to Measure Use of Complementary and Alternative Medicine (I-CAM-Q) and the assessment of the occurrence and factors related to CAM among patients in Poland. A methodological model of translation and cross-cultural adaptation of research tools according to Beaton et al. with a Delphi—Technique was used for the linguistic and cultural adaptation of the Polish version of I-CAM-Q. The Delphi consensus was achieved in the scale assessment between the experts in two rounds (with a score of above 80% of expert agreement). Data was collected using an online survey within 38 thematically different groups on Facebook, among 524 participants. Over half (59.7%, *n* = 313) of the respondents used the services of at least one CAM practitioner. On the other hand, 50.8% (*n* = 266) of the respondents declared using physician’s advice. The use of herbs and plant products was reported by 84.7% (*n* = 444), and vitamins and minerals by 88.4% (*n* = 463) of respondents. The most commonly used self-help practices among the respondents were relaxation techniques (49.6%), praying for one’s health (43.3%) and meditation (41.2%). The consensual methodology of validating the Polish version of the I-CAM-Q scale allowed for creation of a conceptually and linguistically equivalent tool with the original international instrument. A high frequency of CAM use was found among the respondents. Therefore, strategies should be implemented to improve patient-physician communication on the use of CAM in Poland.

## 1. Introduction

According to the World Health Organization (WHO), complementary and alternative medicine (CAM) encompasses a broad set of health care practices that are not part of a country’s traditional or conventional medicine and are not fully integrated into the prevailing health care system [1]. By contrast with the dominant conventional medicine in modern medical science, which hasits origins in scientific knowledge, skills and practices based on theories, beliefs and experiences indigenous to different cultures, CAM medicine encompasses a variety of alternative treatments that have historical roots outside of mainstream medical and are or may be used in conjunction with conventional medicine [2,3]. The use of CAM therapies, for example, acupuncture, herbal medicine, homeopathy, chiropractic and others, is becoming more and more popular in Western societies varied on an individual and national level [2,4].

A large European study [5] has showed that 25.9% of Europeans had used CAM; Poles were in a second to last place, ahead of Hungarians; among the countries participating in the study in terms of CAM use in the general population. The frequency of using CAM among Polish patients ranges from 16% to 85% [6,7,8,9,10,11,12]. However, most studies were conducted on small populations in one centre, among people diagnosed with cancer and using non-validated proprietary research tools. Moreover, Jędrzejewska et al. [10] showed that as many as 71% of Polish patients fail to inform their physicians about the use of CAM. On the other hand, the results of studies by Olchowska–Kotała et al. [13] conducted among physicians in Poland revealed that although physicians are cautious and sceptical regarding CAM methods, they are open to researching CAM and discussing CAM possibilities with their patients.

Due to existence of the inconsistent results describing the percentage of patients using CAM, further studies in a larger population are required. In addition, the high variability of results regarding the use of CAM may be partly explained by the inconsistent definition of CAM, with some authors of research reports considering only herbal remedies. At the same time, others also consider dietary supplements and self-help practices such as meditation or yoga [6,10]. Another aspect that may impact the high variability of the results is the different approach to the form of questionnaires. Some contain open-ended questions, while others use a list of CAM products and practices [11,12]. However, another example that makes a comparison between the results of studies difficult is the time-frame of use CAM. Some researchers ask if the respondent has “ever” used CAM, which often results in inflated rates as respondents indicate that they have used a particular CAM method or product at any time in their lifetime [10,12].

Although there is a growing body of literature documenting patient use of CAM, it is difficult to compare study results due to differences in how CAM use is measured. This also justifies the need to use measurement tools to apply CAM which are recognized by international bodies of CAM experts. The international questionnaire for measuring the use of complementary and alternative medicine (I-CAM-Q) was developed to address the problems associated with the assessment concerning the use of CAM [14]. I-CAM-Q includes a comprehensive and well-thought-out set of items, measuring a wide range of CAM products and practices. The I-CAM-Q has been translated and adapted to many different countries and languages. The questionnaire has been applied worldwide, such as in France [15], Norway [16,17,18,19], Germany [20], Netherlands [21], Sweden [22], USA [23,24], Argentina [25], Japan [26,27,28], Taiwan [29], Korea [30], Iran [31,32], Saudi Arabia [33] and Cambodia [34].

To address the above issues, this study was designed to: (1) provide a linguistic and cross-cultural adaptation of the Polish version of the international questionnaire (I-CAM-Q); and (2) assess the prevalence and factors related to the use of CAM medicine among respondents using a translated and adapted Polish version of the standard international questionnaire I-CAM-Q.

## 2. Materials and Methods

### 2.1. Translation and Cross-Cultural Adaptation of the International Questionnaire I-CAM-Q

For the cross-cultural adaptation, we used a modified version of the method by Beaton et al. [35] with a Delphi technique employing the expert panel review. In the procedure, we considered the general WHO guidelines [36] regarding the process of translating and adapting instruments and the proposal described by Beaton et al. [35], which is why we used the Delphi technique after translation into Polish and back-translation from Polish to English.

After receiving permission from Sara A. Quandt (squandt@wakehealth.edu, e-mail; 2 December 2020), we conducted a linguistic and cultural adaptation of the I-CAM-Q questionnaire in Poland. The cross-cultural adaptation was performed according to the following seven steps: (1) translation (two translators), (2) synthesis (3) panel review according to the Delphi technique (4) back-translation (two back-translators), (5) synthesis, (6) panel review according to the Delphi technique (7) pilot study (Figure 1)

#### 2.1.1. Forward Translation—Step 1

In accordance with the recommendations of Sousa and Rojjanasrirat [37], two (FT1, FT2) translators (a physician and a physiotherapist) translated all components of the I-CAM-Q questionnaire independently. The translators were proficient in the original (source) language (English) and were native speakers of the target language (Polish). The results were reviewed by a person with research experience in constructing and validating the scale, who was proficient in English and whose native language was Polish. As a result of the first stage, two Polish versions of the original I-CAM-Q scale were obtained and subjected to an initial review to compare the received questionnaires in Polish.Each translator produced a written report of the translation.

#### 2.1.2. Comparison and Synthesis of the Polish Translation—Step 2

The two translators and the author of the adaptation (the reviewer) prepared the synthesis the translations results.

#### 2.1.3. Consensus Harmonized Translation and the Cross-Cultural Adaptation—Step 3

The analysis of versions obtained from the translators and the reviewer (the author of the adaptation), revealed four items of the I-CAM-Q scale that needed to be defined more precisely. The version obtained in the first stage of the project cannot be limited to a simple translation, so according to guidelines for the translation and adaptation of tools created by WHO [36], in the next step, the goal was to find the conceptual and semantic equivalents for words or phrases and not a literal translation.

To assess the face and content validity of the questionnaire, consultations with experts were conducted using the Delphitechnique [38]. Implementing the Delphi technique [36] in the linguistic and cultural adaptation of the questionnaire, which is a form of expert research, allowed for obtaining more unambiguous professional judgments in the procedure of proceeding into an authorized and useful shape of the final version of the tool. A panel of experts was composed of ten members: two physicians (an oncologist and an internist), two clinical practice nurses (in oncology and in internal medicine), one physiotherapist, one phytotherapy herbalist, and two scientific researchers, including a reviewer (the author of the adaptation), and forward translators. The experts received the electronic version of the first Polish adaptation of the I-CAM-Q questionnaire along with the following introductory information: "After reading the questions of the I-CAM-Q questionnaire, I would like to request your opinion concerning the understanding of their content. Please use a Likert scale from 1 to 4”. The criteria according to the 4-point Likert scale:(1)—this item is not apparent; (2)—this item requires significant changes to be clear; (3)—this item requires minor changes to be clear; (4)—this item is straightforward. The experts’ responses were then synthesized, assuming that consensus, i.e., the agreement among the experts, would be defined as obtaining an 80% agreement rate. While aiming at a high agreement rate percentage allows for securing the consensus among the experts, it may result in the need to add several consultation rounds. Therefore, the following principles were applied to monitor the consensus:(a)If 80% of the experts rate an item as 4 on the 4-point Likert scale, we consider that there is a consensus on that item and it will be retained in the questionnaire;(b)If 80% of the experts rate an item as 1 on the 4-point Likert scale, we consider it to be no consensus on that item and it will be removed from the questionnaire;(c)For items that were mostly rated as 2 or 3 on the 4-point Likert scale, further discussions are required regarding such items in the questionnaire.

All items were thoroughly reviewed and discussed. Changes were made after discussion and a final synthesized version was agreed.

#### 2.1.4. Back-Translation—Step 4

In the next stage, after the I-CAM-Q questionnaire was deemed acceptable by the panel of experts, third (BT3) and fourth (BT4) translators performed a back translation into English. The first translator, a native English speaker, was from the United States, spoke Polish as the second language and was a professional (English-Polish) language teacher with a master’s degree in education and extensive management, translation and tool adaptation experience. The other translator, a US native whose native language was English, spoke Polish as a second language and had experience in international translations.

#### 2.1.5. Comparison and Synthesis of Back Translation—Step 5

Based on the two back-translated versions, a unified version was established. During the meeting to reach an agreement of experts, it was assessed for conceptual similarity and equivalence and compared with the accepted version versus the original.

#### 2.1.6. Arriving at the Experts’ Consensus—Step 6

The second round of consultations (reconciliations) and ratings were conducted to reach a complete consensus, requiring expert ratings’ agreement of at least 80%. The results of the expert consensus during the 2nd round of the Delphi technique received the average percentage points of agreement at the level of 95.16%. Appendix A presents detailed results of that stage. Ultimately, following the second round of consultations (reconciliations) and ratings, the experts’ consensus was reached and the second, acceptable version of the Polish scale was obtained.

#### 2.1.7. Pilot Study—Step 7

The Polish version of the scale, including the complete cultural adaptation accepted by experts, also includes an understanding of the version obtained by the target population; therefore, phase two was carried out to assess the apparent validity of the scale. First, a pilot study was conducted among thirty oncology patients had no prior medical education to determine the validity of the questionnaire. As it was difficult for the respondents to determine how often they used self-help techniques, the following answers were used: several times, often, very often, every day. At a later stage, a panel of experts, composed of a physician, a physiotherapist, a nurse and a scientific research worker, discussed the results obtained during the implementation of the research using a practice-proven version of the I-CAM-Q questionnaire and drafted a consensus version in Polish, including changes and adaptations stemming from the results obtained in the pilot test, was drafted.

### 2.2. A Cross-Sectional Study

#### 2.2.1. Study Design and Participants

Cross-sectional studies were conducted from 25 June to 18 August 2021. Due to the introduction and continuation of restrictions related to the COVID-19 pandemic, it was decided to use the computer-assisted web interviewing (CAWI) method. The CAWI method made it possible to distribute the questionnaires under conditions of social isolation in a simple and straightforward way. An online survey was conducted, which was available on Facebook for 38 thematically differentiated groups, to obtain the most unified sample possible. The list of thematic groups on Facebook is available in Appendix A.

In order to select groups, a new account was created on the Facebook platform after deleting all browsing history and cookies from the web browser (Google Chrome), then separately entering in the Facebook’s search engine the group category terms related to the most common diseases (e.g., cardiovascular diseases, cancers, intestinal diseases, metabolic diseases) and terms associated with CAM (alternative medicine, unconventional medicine, natural therapies, herbal medicine, Chinese medicine, homeopathy, Ayurvedic medicine).

After entering each term in the search box, the authors joined the first five groups emerging in the Facebook browser and sent a request to the group administrator to post a link to the poll on the main group page. In the event of the group administrator’s disagreement related to adding the account to the group or sharing a post with a link to the survey, the next group found in the Facebook browser was selected to join and the procedure was repeated.

The questionnaire was placed on the Google Forms platform and was made available twice during the research, i.e., on the first and the fourteenth day of the study. In addition, the post with the survey was not deleted and was available for the entire study period. The study was nationwide, and anyone interested could have participated in the study. Participants were allowed to complete the survey only once. Completing online questionnaires is an established method in the health care research field [39]. There is growing evidence that suggests that Facebook is a valuable recruitment tool [40]. This study complies with the STROBE Statement aimed at Strengthening the Reporting of Observational Studies in Epidemiology guidelines and the required information is presented correctly. The STROBE statement checklist for this study is available in Appendix A.

#### 2.2.2. Study Questionnaire

The original I-CAM-Q questionnaire was developed at an international workshop sponsored by the National Research Centre in Complementary and Alternative Medicine (NAFKAM) at the University of Tromsø, Norway [14]. The I-CAM-Q (in Polish adaptation I-CAM-PL) questionnaire has four sections. Part one is concerned with using the services of selected unconventional healthcare providers, such as physicians, chiropractors or herbalists. Respondents are asked to provide reasons they use the selected services (chronic disease, acute disease, well-being or “other”) and how they evaluate their usefulness. In addition, the questionnaire enables the addition of a specified option describing a locally available CAM practice appropriate for the studied population, e.g., *cuarandero* or *cuarander* in the case of Latin American populations in the US and Mexico border states, *Heilpraktiker* in the case of the German population or *Znachor* or *Szeptucha* in the case of the Polish population.

Part two is intended to obtain information on the CAM treatment received from the professionals mentioned above. Part three includes questions about using herbal medicines and dietary supplements, including tablets, capsules and liquids. Study participants are asked to list the products they have used in last 12 months in each of four categories: (1) herbs, (2) vitamins and minerals, (3) homeopathic remedies, and (4) others. Respondents are requested to provide the main reason behind using certain products and how helpful they are to them. The final, fourth part of the questionnaire covers “self-help practices”. The respondents are asked if they have used meditation, yoga or praying for their health in the last 12 months and the main reason for using these practices [14].

Each of the four sections allows respondents to identify different products, practices, or services in the ‘other’ field.

Sociodemographic questions were added to the questionnaire. They included age, gender, place of residence, marital status, professional situation, and financial situation. In addition, the sociodemographic part was supplemented addressing currently treated diseases, the time since the first diagnosis of the disease, and questions regarding the sources of knowledge about CAM.

#### 2.2.3. Ethical Considerations

The Bioethics Committee has issued the Ethical Approval at the Medical University of Lublin (decision number: KE-0254/73/2020). The research was conducted in accordance with the ethical principles contained in Recommendations from the Association of Internet Researchers [41]. Participation in the study was voluntary and anonymous. All study participants gave their informed consent to participate in the survey electronically. The informed consent page explained the purpose and method of the study. There were two options: choosing whether to participate in the study by selecting “Yes” or withdrawing from the study by selecting the “No” option. Only those who chose “Yes” were transferred to the questionnaire page. The respondent could withdraw from the survey at any time by closing the website page containing the questionnaire. The anonymous nature of the online survey made it impossible to track sensitive personal data. After completion, each questionnaire was submitted to the survey platform and the final database was downloaded.

#### 2.2.4. Statistical Analysis

Categorical data are presented as absolute and relative frequencies, and numerical data is presented as a mean with a standard deviation (SD). The Kolmogorov-Smirnov test was used to verify the normality of the data distribution. In order to determine the predictors of using the services of individual groups of specialists, a logistic regression analysis was performed. A backward (conditional) selection method was used. The predictive value of the models was assessed based on the chi-square test (The Omnibus Tests of Model Coefficients) and the Hosmer-Lemeshow goodness of fit test, and Nagelkerke’s R^2^ was also calculated. *p* values < 0.05 was considered statistically significant. All statistical analyses were performed using IBM Corp. (released in 2019) and IBM SPSS Statistics for Windows, Version 26.0. (IBM Corp, Armonk, NY, USA).

## 3. Results

### 3.1. Results of the Linguistic Validation and Cross-Cultural Adaptation of the Polish Version of the I-CAM-Q Scale

The consensus in assessing the Polish version of the I-CAM-Q scale among the experts was arrived at using the Delphi method in two rounds. The detailed results evaluating the validity and clarity of the content of the individual items of the scale are presented in Appendix A.

In the round one, the most significant differences were found in items: 1, 1.3, 1.7 and 3.3., as for those items, the expected ratings’ agreement of 80% was not achieved. The ratings’ agreement range for the individual items came in at: 1–10%, 1.3–60%; 1.7–70% and 3.3–30%.

The cross-cultural adaptation of individual words and phrases included the following determinations:It was impossible to translate the term “health care provider” directly because the term “provider” is not associated with health in Polish. Ultimately, the term was translated as “practitioner”. The professions that are involved in the provision of CAM services are assigned to the group of associates (middle level) health professionals (3230 practitioners of unconventional or complementary methods of therapy), and none of these professions is considered a medical profession [42]. Therefore, the practitioners were divided into medical practitioners in CAM (physician) and non-medical practitioners (others).The proposal was made to add the terms *”Znachor”* and *“Szeptucha”*(whisperer/quack/folk healer) as a general group of people with no medical education who treat people based on folk knowledge.However, in the part related to the therapies proposed by a physician, the “cupping therapy” option was added.The terms “spiritual healer”, “spiritual healing”, and “participation in a traditional healing ceremony” can be understood in Polish as terms related to religious as well as other spiritual practices.

The verification of the items completed, taking into account their conceptual and semantic equivalents, allowed for a complete agreement expressed in the experts’ consensus obtained, including the fulfilment of the ratings’ agreement assumptions for individual items: 1–90%, 1.3–90%; 1.7–90% and 3.3–90%.

### 3.2. Test Results Using the Validated Polish Version of the I-CAM-Q (I-CAM-PL) Questionnaire

#### 3.2.1. Characteristics of the Study Group

Table 1 presents the characteristics of the studied groups. Among 530 collected questionnaires, 524 were completed correctly. Urban inhabitants accounted for 78.6% of the respondents. Respondents were aged between 18 and 74, with a mean of 40.2 ± 11.26. Most respondents (89.7%) were women and people in a relationship (83.3%). Four hundred and four people (77.1%) had higher education, and 78.8% of the respondents were employed. With regard to health, the largest group were people with diagnosed endocrine, nutritional and metabolic diseases (36.1%) and those without any diagnosed disease (16.6%). Two hundred and twenty-five respondents (42.9%) were more than five years since diagnosis.

#### 3.2.2. The Use of CAM among the Respondents

Table 2 presents the results of frequency, motivation and evaluation of CAM use in the study group. The number of respondents who visited a physician taking CAM last year was 266 (50.8%). The respondents most often visited a physician due to a chronic disease (57.5%). In the case of acute disease, 16.2%used the services of a physician. In comparison, the remaining respondents specified the following reasons for visiting a physician: control visit/prophylaxis 10.7%, diagnostic goals 5.3% or a prescription or referral 3.7%. Regarding the remaining CAM therapists, the respondents most often used the services of a herbalist/phytotherapist (27.3%) and an acupuncturist (20.8%). In the opinion of the respondents, among the specialists mentioned, the most useful were visits to a spiritual healer, homeopath and acupuncturist. During the last three months, the respondents most often used the services of an acupuncturist (3.48 ± 4.69) and a physician (2.2 ± 2.96).

Among the CAM methods recommended by the CAM physician, the respondents indicated that they most often used manipulation, followed in frequency by herbs and cupping therapy. The most common reason why physicians recommended CAM therapy (apart from spiritual healing and cupping) was related to chronic diseases. According to the respondents, the most useful methods of CAM therapy were: spiritual healing, acupuncture and herbal medicine. However, a spiritual healing recommended by physicians was declared by a small number of respondents.

The use of vitamins and minerals was declared by 88.4% (*n* = 463) of the respondents and a similar number of respondents declared the use of herbs and herbal medicine (84.7%, *n* = 444). The lowest number of respondents used homeopathic medicine. Interestingly, in the case of the use of herbs and herbal medicine, the respondents indicated chronic disease as the reason for their decision, while in the case of using vitamins and minerals, they most often pointed to desired improvement in well-being, and in the case of homeopathic medicine—Healing of the acute illness. The polls most frequentlymentioned the use of herbs and herbal medicine, vitamins, minerals and homeopathic medicine as very useful. The Appendix A present the respondents’ most commonly used herbal medicinal products and the most frequently used vitamins and minerals. Regarding single herbs, the respondents most often used nettle (*Utrica*), lemon balm (*Melissa*), ashwagandha (*Withania Somnifera*), mint (*Mentha*), cannabidiol oil, white mulberry (*Morus alba*), milk thistle (*Silybum Marianum*) and chamomile (*Matricaria Chamomilla*). On the other hand, among vitamins and minerals, the respondents most often indicated that they use: vitamin D3, vitamin C, magnesium, and vitamin B complex.

Among self-help practices, the respondents most often used relaxation techniques (49.6%, *n* = 260), praying for own health (43.3%, *n* = 227) or meditation (41.2%, *n* = 216), while the least frequently used methods were: attended traditional healing ceremony (3.1%, *n* = 16), Tai Chi (4.4%, *n* = 23) and Qigong (5.9%, *n* = 31). The most common reason the respondents chose self-help practices was to improve their well-being, and most of them admitted that self-help practices are beneficial.

#### 3.2.3. Relationship between Selected Variables and Use of Practitioners of Complementary and Alternative Medicine

Table 3 shows the relationship between sociodemographic variables and the use of complementary and alternative medicine practitioners in the study group. A logistic regression analysis was performed in order to determine the predictors conditioning the use of the services of particular groups of specialists. As a result, only statistically significant predictors are presented in Table 3. The variables that significantly influenced the use of complementary and alternative medicine practitioners turned out to be: age, gender, education, employment status, financial situation and duration of chronic disease.

In the case of using the services of a CAM physician, the probability of making such an appointment drops significantly in the case of people who assessed their financial situation as a good one compared to the group of respondents with a bad financial situation. When it comes to using the services of a chiropractor and a herbalist/phytotherapist, it was observed in the study group that the probability of making such an appointment significantly decreased among people younger 30 years old compared to those aged 40 and over and in patients with a chronic disease up to one year compared from the diagnosis, compared to four or more years in the respondents. Moreover, it was observed that women and the people who are employed more often use the services of herbalists/phytotherapists. In the case of a homeopath, age and gender turned out to be significant predictors. The likelihood of using the services of a homeopath is significantly lower in the age group of up to 30 years compared with people over 40. In turn, a higher probability of using a homeopath’s services is associated with the female gender.

Regarding acupuncturist services, the variables significantly associated with such services were age, duration of chronic disease, and gender. The probability of using the services of an acupuncturist is considerably lower in the age group up to 30 years old compared to people over 40 years old and in patients suffering from up to 1 year compared to people suffering from four years and longer. Women and the people who are employed noted higher probabilities of using the services of an acupuncturist. In the study group, it was observed that the likelihood of using the services of a spiritual healer were significantly lower in patients who were ill for up to one year compared to those who were sick for four years and longer. The higher probability of using the services of a spiritual healer was related to the female gender. As for the likelihood of using the services of a *Znachor/Szeptucha*, the probability of making such an appointment was related to education. More often, such visits are made by people with primary and vocational education than by respondents with higher education.

## 4. Discussion

The international I-CAM-Q questionnaire can be easily customized as it is not specific to any country or culture. Therefore, it is effective and practical in quantifying CAM use. The questionnaire has been translated and adapted for use, in countries such as Iran [32], Germany [20], Taiwan [29] and France [15]. With the help of methodologically correct translation and the process of intercultural adaptation, we were able to transform the published international CAM measurement questionnaire (I-CAM-Q) into the Polish version of the measuring instrument I-CAM-PL. I-CAM-Q includes a comprehensive and well-thought-out set of items that measure a wide range of complementary, alternative and integrative products and practices. Still, it is descriptive, making it impossible to provide a quantitative indicator of CAM use. Therefore, it is impossible to assess the questionnaire Field’s psychometric properties [43].

This study showed that each respondent (100%) used at least one CAM method. Earlier studies conducted in Poland showed the use of CAM in a wide range from 16% to 85% [6,7,8,9,10,11,12]. However, they were mainly carried out among people with cancer and using various questionnaires containing different CAM modalities. Moreover, no studies on the use of CAM in the general population in Poland have been conducted so far.

As in previous studies, the respondents most often used natural products such as herbs or vitamins [12,16,32]. In own research, the use of herbs and plant products was declared by 84.7% of responders. The herbal products most frequently used by respondents include nettle (3.8%), lemon balm (3.4%) and ashwagandha (2.7%). In the case of vitamins and minerals, their use was declared by 88.4% of the respondents. This category’s most frequently used subtypes were vitamin D (20.0%) and vitamin C (11.8%). The most commonly used body and mind practices include relaxation techniques (49.6%), prayer (43.3%) and meditation (41.2%).These results align with the findings of previous studies among Polish patients with the neoplastic disease [10]. Prayer for one’s health is a frequently used CAM method, regardless of the culture and religion [31,32,44,45]. The authors of previous studies on the use of CAM by Polish patients only included self-help products and/or techniques in the questionnaires. This study is the first to consider the use of CAM practitioners. Using the services of at least one CAM practitioner (excluding a physician) was declared by more than half (59.7%) of the respondents, while the services of a physician were reported by 50.8%.

The majority of the CAM methods used by the participants in this study were rated helpful/very helpful. These results are consistent with the findings of researchers from Norway, Sweden and Germany [20,22,26]. On the other hand, studies by Kasprzycka et al. [12] among cancer patients indicate that most respondents did not experience any health benefits of CAM use.

The probability of using the services of a chiropractor, herbalist, homeopath, acupuncturist, or another CAM practitioner significantly decreases in the age group up to 30 years old (compared to people over 40) and who have been four years or more since the diagnosis. According to previous studies, gender is statistically significantly associated with the use of CAM [2]. Being of the female gender was an essential determinant in having a higher probability of using the services of a herbalist, acupuncturist, spiritual healer, and homeopath. Previous studies have shown that education, employment and financial status are related to the use of CAM [2,5]. The probability of using the services of a herbalist and acupuncturist is higher among people who are employed. The level of education determined the use of the *znachor/szeptucha* services and other CAM practitioners specified by the respondent. People with only primary education more often used the services of a *znachor/szeptucha*, while among respondents with higher education, the probability of using “other” CAM practitioners increases. It has been suggested that those with a higher socioeconomic position may be willing to choose and control their approach to health issues better. However, it was noted that although CAM users may be better educated, this does not necessarily mean that they are better informed about the effectiveness of alternative treatments [46]. Our research showed that a bad financial situation was associated with more frequent visits to a physician.

The very high percentage of CAM use in this study is probably due to the inclusion of herbal medicines, dietary supplements and self-health prayers as the CAM modality. Prayer is not commonly regarded as a method of alternative medicine in Poland. In addition, the online version of the I-CAM-Q questionnaire may contribute to fewer missing data compared to the paper-pencil version [20]. Undoubtedly, the lifestyle changes and blocking measures resulting from the SARS-CoV-2 pandemic could have moved patients away from formal health care facilities toward CAM practitioners who met their needs and expectations [21,47].

Further research using the I-CAM-PL questionnaire should focus on determining which CAM methods are used by Polish patients with specific disease entities and to correlate to most typical diseases for a given group of patients. These activities will enable the assessment of the goodness of CAM practices and products in accordance with the principles of integrative medicine—in terms of their safety and effectiveness.

### The Strengths and Limitations

The strengths of these studies require further consideration: First, to the best of our knowledge, this is the first study to assess CAM use among the general population of Poland. Second, a validated standardized questionnaire, which has not yet been used to evaluate CAM among the Polish population, allows for an international comparison of the results. Third, our research was conducted during the COVID-19 pandemic when blocking measures were in place to restrict access to formal healthcare facilities, which likely allows for assessing the impact of blocking measures on respondents’ search for other forms of meeting their health-related needs.

Nevertheless, this study has several limitations. First, a cross-sectional study design enables the inference of correlation, not the construction of causality. Second, the evaluation of the effectiveness of selected CAM services, products and practices is solely the subjective opinion of the respondents. Third, due to the limitations of the COVID-19 pandemic, we have adopted a CAWI research strategy. Therefore, collecting data from people who refused to participate in the study was not possible, and no percentage of refusals was recorded. In addition, during the recruitment process, only the Facebook social network was used, which is why there are no participants in the surveyed population who do not use this portal or are not members of the groups we selected.

Nevertheless, the results of our study and the creation of the Polish version of the I-CAM-Q questionnaire may constitute a valuable reference point for the discussion on the application of CAM.

## 5. Conclusions

In conclusion, we found that using a systematic and consensual methodology allowed us to obtain the Polish version of the I-CAM-Q scale, which is conceptually and linguistically equivalent to the original international instrument and adequate for assessing the use of CAM in the general Polish population. Although it is not possible to determine the psychometric properties of the questionnaire, it is an accurate tool for assessing the use of CAM methods among the Polish-speaking population. Furthermore, there was a high CAM application rate among respondents in Poland.

The main determinants of CAM application are age, gender, relationship status, financial situation, employment situation, place of residence, and duration of illness. Integrating orthodox medicine with CAM methods should be proclaimed by implementing a strategy to improve communication between the patient and the physician about the use of CAM. In addition, national efforts should focus on informing the public about legal CAM benefits and the dangers of using illegal CAM services.

## Figures and Tables

**Figure 1 ijerph-20-00124-f001:**
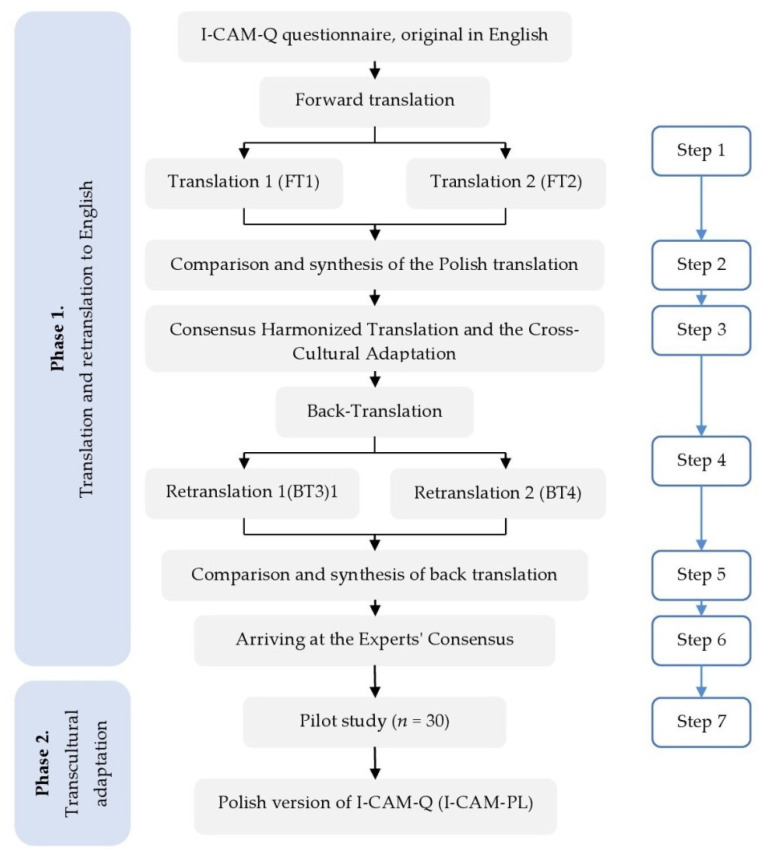
Phases of linguistic validation and the cross-cultural adaptation process of the Polish version of the I-CAM-Q questionnaire.

**Table 1 ijerph-20-00124-t001:** Characteristics of the researched group (*n* = 524).

Variables	*n* (%)
Gender:	
Female	470 (89.7)
Male	54 (10.3)
Age (years):	
≥30	104 (19.8)
31–40	184 (35.1)
≤41	236 (45.0)
Education:	
Elementary	8 (1.5)
Vocational	12 (2.3)
Secondary	100 (19.1)
Higher	404 (77.1)
Employment status:	
Employed	413 (78.8)
Unemployed	111 (21.2)
Financial situation:	
Very good	92 (17.6)
Rather good	245 (46.8)
Average	169 (32.3)
Rather bad	14 (2.7)
Bad	4 (0.8)
Place of living:	
City/town	412 (78.6)
Village	112 (21.4)
Marital status:	
In relationship	436 (83.3)
Single	76 (14.5)
Widowed	12 (2.3)
Type of disease:	
Certain infectious and parasitic diseases	37 (7.1)
Neoplasm	12 (2.3)
Diseases of the blood and blood-forming organs and certain disorders involving the immune mechanism	9 (1.7)
Endocrine, nutritional and metabolic diseases	189 (36.1)
Mental and behavioural disorders	75 (14.3)
Diseases of the nervous system	38 (7.3)
Diseases of the eye and adnexa	5 (1.0)
Diseases of the ear and mastoid process	2 (0.4)
Diseases of the circulatory system	60 (11.5)
Diseases of the digestive system	61 (11.6)
Diseases of the respiratory system	23 (4.4)
Diseases of the skin and subcutaneous tissue	34 (6.5)
Diseases of the musculoskeletal system and connective tissue	83 (15.8)
Diseases of the genitourinary system	27 (5.2)
Symptoms, signs and abnormal clinical and laboratory findings, not elsewhere classified	18 (3.4)
Injury, poisoning and certain other consequences of external causes	43 (8.2)
Unspecified by the respondent	20 (3,8)
No disease	87 (16.6)
Time since the diagnosis:	
0–1 year	73 (13.9)
2–3 years	74 (14.1)
4–5 years	82 (15.6)
More than 5 years	225 (42.9)
Not relevant, no disease	70 (13.4)

**Table 2 ijerph-20-00124-t002:** The frequency, motivation and evaluation of CAM use in the study group during the last 12 months.

Visited/Used ^a, #^		Motivation ^a, ##^					Usefulness of the Visit/Product/Self-Help Technique ^a, ##^					Number of Visits in the Last 3 Months ^b^
		Chronic Disease	Acute Illness	To Improve Well-being	Others	Missing Data	Very	Some What	No at All	Do Not Know	Missing Data	
1. Visiting practitioners of complementary and alternative medicine:												
Physician	266 (50.8)	153 (57.5)	43 (16.2)	0 (0)	48 (18.0)	22 (8.3)	89 (33.5)	103 (38.7)	40 (15.0)	14(5.3)	20 (7.5)	2.20 ± 2.96
Chiropractor	69 (13.2)	34 (49.3)	6 (8.7)	28 (40.6)	0 (0)	1 (1.4)	47 (68.1)	19 (27.5)	0 (0)	2 (2.9)	1 (1.4)	1.86 ± 1.83
Herbalist/Phytotherapist	143 (27.3)	79 (55.2)	8 (5.6)	55 (38.5)	0 (0)	1 (0.7)	107 (74.8)	26 (18.2)	4 (2.8)	4 (2.8)	2 (1.4)	1.58 ± 1.37
Homeopath	79 (15.1)	32 (40.5)	22 (27.8)	17 (21.5)	5 (6.3)	3 (3.8)	62 (78.5)	12 (15.2)	3 (3.8)	1 (1.3)	1 (1.3)	1.1 ± 1.06
Acupuncturist	109 (20.8)	55 (50.5)	13 (11.9)	35 (32.1)	5 (4.6)	1 (0.9)	82 (75.2)	21 (19.3)	3 (2.8)	2 (1.8)	1 (0.9)	3.48 ± 4.69
Spiritual healer	85 (16.2)	25 (29.4)	1 (1.2)	51 (60.0)	8 (9.4)	0 (0)	69 (81.2)	16 (18.8)	0 (0)	0 (0)	0 (0)	2.06 ± 3.14
*Znachor/Szeptucha*	19 (3.6)	5 (26.3)	2 (10.5)	8 (42.1)	2 (10.5)	2 (10.5)	14 (73.7)	2 (10.5)	2 (10.5)	1 (5.3)	0 (0)	1.39 ± 1.09
Other specialist ^¥^	132 (25.2)	64 (48.5)	9 (6.8)	50 (37.9)	6 (4.5)	3 (2.3)	108 (81.8)	19 (14.4)	3 (2.3)	1 (0.8)	1 (0.8)	3.99 ± 2.00
2. Complementary treatments received from physicians:												
Manipulation	142 (27.1)	66 (46.5)	18 (12.7)	51 (35.9)	4 (2.8)	3 (2.1)	88 (62.0)	47 (33.1)	3 (2.1)	2 (1.4)	2 (1.4)	-
Homeopathy	53 (10.1)	28 (52.8)	13 (24.5)	4 (7.5)	4 (7.5)	4 (7.5)	33 (62.3)	14 (26.4)	2 (3.8)	3 (5.7)	1 (1.9)	-
Acupuncture	37 (7.1)	22 (59.5)	8 (21.6)	5 (13.5)	1 (2.7)	1 (2.7)	24 (64.9)	11 (29.7)	1 (2.7)	1 (2.7)	0 (0)	-
Herbs	113 (21.6)	64 (56.6)	20 (17.7)	23 (20.4)	5 (4.4)	1 (0.9)	72 (63.7)	33 (29.2)	2 (1.8)	5 (4.4)	1 (0.9)	-
Spiritual healing	23 (4.4)	7 (30.4)	2 (8.7)	14 (60.9)	0 (0)	0 (0)	16 (70.0)	5 (22.0)	0 (0)	0 (0)	2 (9.0)	-
Cupping therapy	58 (11.1)	19 (32.8)	21 (36.2)	9 (15.5)	3 (5.2)	6 (10.3)	34 (58.6)	17 (29.3)	3 (5.2)	1 (1.7)	3 (5.2)	-
Others^¥^	39 (7.4)	25 (64.1)	5 (12.8)	7 (17.9)	2 (5.1)	0 (0)	24 (61.5)	12 (30.8)	0 (0)	3 (7.7)	0 (0)	-
3. Use of herbal medicine and dietary supplements:												
Herbs/herbal medicine	444 (84.7)	424 (41.5)	144 (14.1)	337 (33.0)	92 (9.0)	24 (2.4)	683 (66.9)	250 (24.5)	19 (1.9)	49 (4.8)	20 (2.0)	-
Vitamins/minerals	463 (88.4)	415 (38.2)	58 (5.3)	429 (39.5)	161 (14.8)	22 (2.0)	694 (64.0)	239 (22.0)	11 (1.0)	134 (2.4)	7 (0.6)	-
Homeopathic medicines	100 (19.1)	73 (35.4)	100 (48.5)	21 (10.2)	1 (0.5)	11 (5.3)	164 (79.6)	24 (11.7)	5 (2.4)	10 (4.9)	3 (1.5)	-
Other supplements ^¥^	221 (42.2)	90 (41.9)	19 (8.8)	83 (38.6)	21 (9.8)	2 (0.9)	159 (74.0)	39 (18.1)	3 (1.4)	14 (6.5)	0 (0)	-
4. Self-help practices:												
Meditation	216 (41.2)	36 (16.7)	2 (0.9)	160 (74.1)	17 (7.9)	1 (0.5)	158 (73.1)	52 (24.1)	4 (1.9)	2 (0.9)	0 (0)	-
Yoga	168 (32.1)	33 (19.6)	4 (2.4)	119 (70.8)	10 (6.0)	2 (1.2)	122 (72.6)	35 (20.8)	1 (0.6)	8 (4.8)	2 (1.2)	-
Qigong	31 (5.9)	5 (16.1)	0 (0)	23 (74.2)	3 (9.7)	0 (0)	21 (67.7)	0 (0)	8 (25.8)	2 (6.5)	0 (0)	-
Tai Chi	23 (4.4)	2 (8.7)	0 (0)	21 (91.3)	0 (0)	0 (0)	13 (56.5)	8 (34.8)	0 (0)	2 (8.7)	0 (0)	-
Relaxation techniques	260 (49.6)	39 (15.0)	3 (1.2)	203 (78.1)	13 (5.0)	2 (0.8)	167 (64.2)	78 (30.0)	5 (1.9)	9 (3.5)	1 (0.4)	-
Visualization	185 (35.3)	34 (18.4)	6 (3.2)	122 (65.9)	20 (10.8)	3 (1.6)	115 (62.2)	49 (26.5)	2 (1.1)	18 (9.7)	1 (0.5)	-
Attended traditional healing ceremony	16 (3.1)	5 (31.1)	1 (6.3)	9 (56.3)	1 (6.3)	0 (0)	10 (62.5)	4 (25.0)	0 (0)	2 (12.5)	0 (0)	-
Praying for own health	227 (43.3)	71 (31.3)	14 (6.2)	106 (46.7)	28 (12.3)	8 (3.5)	141 (62.1)	45 (19.8)	3 (1.3)	34 (15.0)	4 (1.8)	-
Others ^¥^	99 (18.9)	23 (23.2)	3 (3.0)	61 (61.6)	11 (11.1)	1 (1.0)	81 (81.8)	13 (13.2)	0 (0)	4 (4.0)	1 (1.0)	-

Data are presented as ^a^
*n* (%); ^b^ mean ± SD; ^#^ percentages are calculated in relation to the total sample population; ^##^ percentages are calculated in relation to the number of people using a given health care provider; ^¥^ due to a large number of different answers, it was not possible to compare.

**Table 3 ijerph-20-00124-t003:** Relationship between selected sociodemographic variables and use of services of complementary and alternative medicine.

Variable	OR	95% CI	*p*
Physician:			
Financial situation (reference category: bad and rather bad)	1		
Average	0.637	[0.637–2.133]	0.463
Very good and rather good	0.376	[0.115–0.926]	0.045
Chiropractor:			
Age (reference category: Over 40 years old):	1		
31–40 years	0.551	[0.290–1.045]	0.068
Up to 30 years	0.341	[0.143–0.818]	0.016
Time since the disease diagnosis (reference category: 4 years and more):	1		
2–3 years	2.485	[1.299–4.122]	0.056
Up to 1 year	0.356	[0.106–0.993]	0.045
Herbalist/Phytotherapist:			
Gender (reference category: male)	1		
Female	3.844	[1.475–10.012]	0.006
Age (reference category: over 40 years old):	1		
31–40 years	0.921	[0.578–1.467]	0.728
Up to 30 years	0.525	[0.284–0.970]	0.04
Employment status (reference category: Unemployed)	1		
Employed	1.784	[1.028–3.099]	0.04
Time since the disease diagnosis (reference category: 4 years and more):	1		
2–3 years	1.418	[0.808–2.490]	0.224
Up to 1 year	0.579	[0.284–0.943]	0.046
Homeopath:			
Gender (reference category: male)	1		
Female	4.228	[1.008–18.034]	0.049
Age (reference category: over 40 years old):	1		
31–40 years	1.439	[0.813–2.545]	0.211
Up to 30 years	0.583	[0.253–0.823]	0.014
Acupuncturist:			
Gender (reference category: male)	1		
Female	2.579	[1.128–6.775]	0.044
Age (reference category: over 40 years old):	1		
31–40 years	1.198	[0.725–1.979]	0.481
Up to 30 years	0.531	[0.261–0.921]	0.039
Employment status (reference category:unemployed)	1		
Employed	1.866	[1.000–3.493]	0.049
Time since the disease diagnosis (reference category: 4 years and more):	1		
2–3 years	1.399	[0.763–2.567]	0.277
Up to 1 year	0.461	[0.208–0.999]	0.05
Spiritual healer:			
Gender (reference category: male)	1		
Female	3.161	[1.161–10.518]	0.041
Time since the disease diagnosis (reference category: 4 years and more):	1		
2–3 years	0.585	[0.264–1.297]	0.187
Up to 1 year	0.327	[0.125–0.853]	0.022
*Znachor/Szeptucha:*			
Education (reference category: higher)	1		
Secondary	3.005	[0.944–9.567]	0.063
Elementary and vocational	6.976	[1.287–37.800]	0.024

## Data Availability

The datasets used and/or analysed during the current study are available from the corresponding author on reasonable request.

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
