# Peer review of "Translation and Cross-Cultural Adaptation of the International Questionnaire to Measure the Use of Complementary and Alternative Medicine (I-CAM-Q) for the Polish and Cross-Sectional Study"

_ijerph, 2022, doi:10.3390/ijerph20010124_

Round 1
Reviewer 1 Report
The paper is well designed and discussed in nteresting way. The article can be published after minor revision to the reference list and add the DOI for the cited articles to be more obvious and easy to be reached with the readers
Reviewer 2 Report
Please finde here the results of reviewing „Translation and Cross-Cultural Adaptation of the International Questionnaire to Measure the Use of Complementary And Al-3 ternative Medicine (I-CAM-Q) for the Polish and Cross-Sectional Study“. Although I very much support the idea of adapting one measurement instrument of CAM use across Europe and the world for providing comparable data I suggest to revise the manuscript thouroughly. The manuscript as it is now represents rather 2 studies than explaining the approach of how the instrument was adapated. I suggest the following:
Introduction
Instead of giving a rather political background highlighting the contrast to the „dominant“ conventional approach, and looking at the coverage of CAM the introduction shoul convey a clear understanding oft he problems of measuring CAM use (e.g. different definitions, and learning points from all I-CAM-Q versions mentioned here).
Methods:
It remains totally obscure in what way the Delphi method informed the cultural adaption. Please give more detail (what was given to the members of the Delphi rounds?, who was among them?, which questions had to be answered, etc.)
Judging from the headings and sub-headings, the Delphi-Process is given as heading, and translation was one part of this process. This has to be made very clear under 2.1.
2.1.1.
Sentence one: what is meant by reviewer here?
Which items were ambigeous? Who rated this? What does the statement of translation and adaption has to do with these four items?
Line 103 – 107
As the questionnaire has to be filled in by patients it is highly unfortunate that no persons from this group (people without medical knowledge and without expert knowledge of CAM) were part of this adapation process. Later in the manuscript patients are mentioned, but it seems that experts delivered the final assessment of validity. It should be made clearer in what way the voice of patients contributed to this.
Unfortunately, Figure 1 does not help to understand the process. The figure has to be streamlined with the details given in 2.1.
Line 113f: Are these the same experts as in 2.1.1. Does that mean that a first version was compiled after 2.1.1.?
Line 114ff: The approaches rather refer to face validity. Content validity needs other methods to be assured.
Line 123: which model?
Line 129ff: It is not clear what is meant by "item". Items are a composite of a question and response options. The I-Cam-Q probably presents some challenges in response options (rather than the question itself). Please be precise.
Line 133 leave out "acceptable" as no method was applied to show any form of validity or reliability in the group that has to answer the items (patients!)
Please check also line 145: acceptable to whom? Was this the 2nd version??
2.1.3
How come that the forward translation was done after a 2nd version was formulated? If this was done before the 2nd version was created please check order and content of the several subheadings
2.1.5.
Apparent validity is unclear. Probably, face validity is meant
Line 206: What is meant by implementation?
All in all, this manuscript should be EITHER focused on the adaption process, and give a clear answer tot he question: in which way had the instrument to be adpated to be used in Polish, and give empirical data for this.
OR the manuscript could focus on CAM use measured by the instrument. In this case there should only be an (additional) paragraph in the method section which explains the items that were adapted (and in what way). The other text should then give methods and results of the survey.
There are some minor language issues, e.g. chiropractors where chiropractice ist meant, and irritating use of "the".
Reviewer 3 Report
letter attached

Round 2
Reviewer 2 Report
Unfortunately, the problems detected and explained in report 1 are all still present. Major problems are: no adherence to standards of language adaption (e.g. forward and backward translation). The delphi method is no acceptable tool, and does not provide scientifiv quality during this adaption process. The introduction is even longer (still bearing a lot of unnecessary details), and the new paragraphs do not contain any references. There are a lot of problems with the English language. The paper did not improve (no merit due to the intransparent and low quality process).
